# Analysis of Volatile Profile and Aromatic Characteristics of Raw Pu-erh Tea during Storage Based on GC-MS and Odor Activity Value

**DOI:** 10.3390/foods12193568

**Published:** 2023-09-26

**Authors:** Jie Guo, Zhihao Yu, Meiyan Liu, Mengdi Guan, Aiyun Shi, Yongdan Hu, Siyu Li, Lunzhao Yi, Dabing Ren

**Affiliations:** 1Faculty of Food Science and Engineering, Kunming University of Science and Technology, Kunming 650500, China; guojie101298@163.com (J.G.); 13324915711@163.com (Z.Y.); lmy13562173653@163.com (M.L.); guan1230414@163.com (M.G.); kunlan881221@126.com (Y.H.); mnblsy@163.com (S.L.); yilunzhao@kust.edu.cn (L.Y.); 2Yunnan TAETEA Group Co., Ltd., Kunming 650500, China; 18787997350@163.com

**Keywords:** raw Pu-erh tea, storage, volatile compounds, HS-SPME-GC-MS, aroma characteristic

## Abstract

Volatile constituents are critical to the flavor of tea, but their changes in raw Pu-erh tea (RAPT) during storage have not been clearly understood. This work aimed to investigate the volatile composition and their changes at various storage durations. The volatile profile of RAPT was determined using headspace solid-phase microextraction in combination gas chromatography–mass spectrometry. A total of 130 volatile compounds were identified in RAPT samples, and 64 of them were shared by all samples. The aroma attributes of RAPT over a storage period ranging from 0 to 10 years were assessed through the combination of odor activity value (OAV), aroma characteristic influence(ACI) value, and multivariate statistical analysis. The results revealed that RAPT exhibited a distinct floral and fruity aroma profile after storage for approximately 3–4 years. A notable shift in aroma was observed after 3–4 years of storage, indicating a significant turning point. Furthermore, the likely notable shift after 10 years of storage may signify the second turning point. According to the odor activity value (OAV ≥ 100), eight key volatile compounds were identified: linalool, α-terpineol, geraniol, trans-β-ionone, α-ionone, (E,E)-2,4-heptadienal, 1-octanol, and octanal. Combining OAV (≥100) and ACI (≥1), five compounds, namely linalool, (E,E)-2,4-heptadienal, (Z)-3-hexen-1-ol, 2,6,10,10-tetramethyl-1-oxaspiro [4.5]dec-6-ene, and octanal, were identified as significant contributors to the aroma. The results offer a scientific foundation and valuable insights for understanding the volatile composition of RAPT and their changes during storage.

## 1. Introduction

Tea has a rich historical background and is highly appreciated by a diverse group of consumers owning to its unique aroma, taste, and biological properties. The principal categories of Chinese tea include green tea, yellow tea, white tea, oolong tea, black tea, dark tea or fermented tea, and Pu-erh tea [1,2]. Pu-erh tea is predominantly produced in Yunnan province of China and utilizes the fresh leaves of Yunnan’s distinctive large-leaf tea (*Camellia sinensis*) cultivar as material [3,4]. Pu-erh tea consists of raw Pu-erh tea (RAPT) and ripe Pu-erh tea [5]. The manufacturing process for RAPT bears resemblance to that of traditional green tea, involving stages such as withering, de-enzyming, rolling, drying, and autoclaving–compressing. It is worth mentioning that while green tea typically has a limited shelf life of 1–2 years, RAPT tea can be stored for an extended period exceeding 5 years. The prolonged storage duration significantly contributes to the alteration and enhancement of the flavor profile of Pu-erh green tea, thus assuming a crucial role.

An abundant and distinct aroma plays a crucial role in determining the quality of tea, and the different aroma compounds contribute to the various sensory characteristics of the tea [6]. In a previous study, over 200 volatile compounds were identified in RAPT, with a predominant presence of alcohols, ketones, aldehydes, and methoxy compounds [7]. Methoxyphenolic compounds (stale/musty aroma notes) and ketones (woody or floral aroma notes) play a vital role in the special flavor of ripened Pu-erh tea. In RAPT, the floral aroma (38.62%) and fruity aroma (16.55%) were the major aroma categories [8]. Moreover, the amounts of alcohols contribute to the floral-like of tea, such as geraniol. The majority of aldehydes reported in tea are known to contribute to the citrus-like and green flavor of tea infusion. Also, some of them are responsible for bitter-almond-like, honey-like, and even fatty odor notes. Ketones constitute the second largest group of odorants that have been reported in tea to date, contributing to flowery, fruity, and woody aroma notes [9]. In summary, RAPT contains a diverse range of aroma compounds, and the composition of these compounds impacts the flavor profile of tea.

Generally, the aroma composition of tea is affected by various factors. Previous studies have investigated the impact of manufacturing process, such as de-enzyming and withering, on the aroma composition of tea. Fan et al. [10] reported that the de-enzyming and autoclaving–compressing treatments significantly changed the volatile composition of RAPT. Storage is also a crucial factor in the change in and formation of tea aroma. Qi et al. [11] found that aged white tea developed a sweet fragrance with herbal aroma, while the fresh white tea displays a fresh, delicate fragrance with mild green aroma. Additionally, the aroma of black tea undergoes significant changes during storage. Meng and co-workers reported that after five years of storage, methoxy benzene volatiles became the storage markers of Keemun congou black tea, contributing to a stale odor [12]. Then, a study revealed the aroma changes during storage of An tea (AT); the results showed that, as the storage time increased, the tea exhibited an increase in stale and woody aromas, for example, 4-ethyl-1,2-dimethoxybenzene (0–2.26 μg/L) and 1,2,3-trimethoxy-5-methyl-benzene (0–14.81 μg/L) [13]. This suggests that storage has a significant impact on the volatile composition and the formation of tea aroma. Nevertheless, it should be noted that there currently are no comprehensive investigations evaluating and comparing the flavor attributes of RAPT during storage, particularly with regard to key aroma compounds and the change in aroma.

In recent years, many fields have benefitted from the advancements in instrumental technology, including the rapid development of food flavor analysis techniques. One such technique involves the combination of gas chromatography (GC) with mass spectrometry (MS), known as GC-MS, which has been widely used for odor analysis and quality classification in food, especially in relation to tea [14,15,16,17]. Sample extraction is a crucial step in the analysis of volatile components using GC-MS, with headspace solid phase microextraction (HS-SPME) being one of the prevailing methods employed for this purpose. Nowadays, HS-SPME combined with GC-MS (HS-SPME-GC-MS) has been widely used for the determination of volatile compounds in various food products due to its high sensitivity and good reproducibility [18,19,20]. In a recent study, HS–SPME–GC–MS was employed to comprehensively compare and analyze the flavor-active compounds in GABA white tea processed by natural withering and combined withering. The results showed that combined withering was more favorable for the formation of floral and sour-fruity aroma compared with natural withering. It was also revealed that 2-heptanol plays an important role in contributing to the aroma of GABA white tea [21].

This study aimed to comprehensively analyze the volatile composition of RAPT and to investigate its changes during storage, considering the significant effects of storage on its aroma. To achieve this, an efficient HS-SPME–GC–MS method was used to extract and detect the volatile constituents in RAPT samples collected at different storage stages. The obtained GC–MS data were processed and interpreted using various chemometric methods, including ANOVA, PCA, HCA, and heatmap analysis. By combining OAV and ACI values, the key volatile compounds and major aroma-contributing compounds were identified, providing insights into the influence of volatile component changes on the aroma characteristics of RAPT at different storage times. These findings will contribute to a better understanding of the volatile composition and the unique aroma of RAPT during storage.

## 2. Materials and Methods

### 2.1. Chemicals and Reagents

De-ionized water was produced using a Milli-Q purification system (Milli-Q A10, Merck, Darmstadt, Germany). N-alkanes were obtained from o2si smart solutions Corporation (chromatographic pure grade, C8–C40). Sodium chloride (NaCl) was obtained from Innochem (Beijing, China). Decanoic acid ethyl ester was obtained from Sigma-Aldrich (purity of 98%).

### 2.2. Tea Samples

In this work, tea samples aged between 0 and 10 years were utilized. The samples denoted as T0, T1, T2, T3, T4, T5, T6, T7, T8, T9, and T10 were obtained from the products of the RAPT 7542 series (Yunnan TAETEA Group Co., Kunming, China), which exhibits a cake-like morphology. All samples were derived from the same tea variety, cultivated at the same location (Xishuangbanna, Yunnan, China), harvested during the same season (Spring tea), and processed using identical methodologies. Following production, the samples were stored in Guangdong Province. Prior to analysis, 100 g of each tea sample was passed through a 60 mesh sieve to obtain the tea powder and subsequently refrigerated at −20 °C for future use.

### 2.3. Extraction of Volatile Compounds by HS-SPME

The volatile compounds of RAPT were extracted by HS-SPME. Firstly, volatile compounds were extracted and enriched from RAPT samples using a 50/30 μm DVB/CAR/PDMS fiber head. In order to prevent contamination, the extracted fibers needed to be maintained at 270 °C for 1 hour in the GC-MS inlet before use.

For the HS-SPME process, a tea powder/water ratio (1:10 g/mL) was used. The extraction temperature was set at 80 °C, and the extraction time was 40 min. Precisely weighed, 0.200 g of tea powder was added into a 10 mL headspace vial, followed by the addition of 1.8 g of NaCl and 10 μL of ethyl decanoate (0.04 mg/mL) as an internal standard. Subsequently, 2 mL of boiling water was added, and the headspace vial was immediately sealed. The extracted fibers were then inserted into the vial and kept at 80 °C for 40 min. Once the extraction was complete, the SPME fiber needle was removed and inserted into the GC injection port for desorption, which lasted for 5 min at 260 °C.

### 2.4. Volatile Component Profiling via GC–MS

GC-MS (QP2010 Shimadzu, Kyoto, Japan) and an HP-5MS UI quartz capillary column (30 m × 0.25 mm × 0.50 μm) were employed in this experiment. The inlet temperature was 260 °C. The carrier gas was high-quality helium (>99.999%) at a flow rate of 1 mL/min; the sample was injected by split flow with a 5:1 split ratio. The column temperature was 50 °C and increased to 80 °C at a rate of 10 °C/min. Then, the temperature was increased to 90 °C at a rate of 3 °C/min, 90 °C for 3 min and heated up to 120 °C at a rate of 3 °C/min, 120 °C for 3 min, which then continued at 3 °C/min to 170 °C. Finally, the temperature was increased to 230 °C at a rate of 15 °C/min and held for 4 min.

Mass spectrometry (MS) conditions: The ion source was an EI source with an electron ionization energy of 70 eV and a scan range of *m*/*z* 30–540 amu. The temperature of the ion source was 230 °C, and the temperature of the interface was 260 °C. The solvent delay time was set to 3.2 min. 

### 2.5. Identification and Annotation of Volatile Compounds

The data were screened from the database in the system for compounds with >80% similarity, and retention indices (*RIs*) were determined using a mixture of n-alkanes (C8-C40). The *RIs* were calculated as follows:(1)$$RIs=100n+\frac{100\left(t_{i}-t_{n}\right)}{t_{n+1}-t_{n}}$$
where *t_i_* is the retention time of the compound to be measured; *n* is the number of carbon atoms in the n-alkane mixture; and *t_n_* and *t_n+_*_1_ are the retention times of the n-alkane standard mixture with *n* and *n* + 1 carbon atoms, respectively (*t_n_* < *t_i_* < *t_n_*_+1_) [22].

After that, the volatile compounds were matched and annotated using the National Library of Standards and Technology (NIST) spectral library (https://webbook.nist.gov/chemistry/name-ser/ (accessed on 1 March 2023)). 

The content of each volatile compound was determined using the internal standard method and was calculated based on the following equation. Here, decanoic acid ethyl ester (0.04 mg/mL) was used as the internal standard compound.
(2)$$Y=\frac{ACV}{mA_{IS}}\times 100000$$
where *Y* represents the content of each volatile compound (μg/100 g); *C* is the concentration of the internal standard (mg/mL); *V* denotes the volume of the internal standard (mL); *m* is the mass of the added sample (g); and *A* and *A_IS_* represent the peak areas of the volatile compounds and the internal standard, respectively [10].

The relative content of each volatile component was calculated as follows: Relative content (%) = single component content/total content × 100% [10].

### 2.6. Calculation of Odor Activity Value (OAV) and Aroma Character Impact (ACI)

The odor activity value (*OAV*) represents the ratio of the dilution concentration of volatile compounds in water to its threshold value, which can better represent the degree of contribution of volatile compounds to the aroma [23,24], and the *OAV* value is calculated as follows:(3)$$OAV=\frac{C}{OT}$$
where *C* is the concentration of the compound in the sample (μg/mL) and *OT* is the odor threshold of the compound in water (μg/mL) [25].

In addition to analyzing the contribution of a single compound to the aroma in RAPT, the contribution of a mixture of compounds in RAPT to the aroma is also analyzed, i.e., the relative degree of contribution of each component in the mixture to the aroma is analyzed using the aroma characteristic influence (*ACI*) value, calculated as follows:(4)$$ACI(\%)=\frac{C_{i}T_{i}}{\sum_{k}C_{k}T_{k}}\times 100=\frac{O_{i}}{\sum_{k}O_{k}}\times 100$$
where *C_i_* denotes the concentration of a given compound (*i*), *T_i_* is the odor threshold of the compound in water reported in the literature, *O_i_* is the *OAV* value of compound (*i*), and *O_k_* is the *OAV* value of any compound [26].

### 2.7. Statistical Analysis

All experiments were carried out independently, and the results were expressed as mean plus standard deviation (*n* = 3). The software SPSS 25 (IBM Inc., Chicago, IL, USA) was used to determine significant differences (*p* < 0.05) for volatile compounds. The flavor characteristics of volatile compounds in RAPT were determined based on the data reported in the literature and on websites and were used for significance analysis. The used plotting software included Origin Pro version 2023 and an online plotting webpage (https://www.chiplot.online (accessed on 1 June 2023)).

## 3. Results and Discussions

### 3.1. Volatile Profile of RAPT with Various Years of Storage

In the present work, an HS-SPME-GC-MS method was optimized and developed to profile the volatile compounds in RAPT with different storage durations (1–10 years). Based on the optimized method, qualitative and quantitative analyses of the volatile compounds were performed. As illustrated in Appendix A (total ion chromatograms (TIC)), a large number of volatile compounds were detected in RAPT, and the profile of RAPT with different years of storage varied from each other. The data analysis was performed using a commercial software (GSMSsolution, version 4.30, labSolition, Shimadzu, Kyoto, Japan). A total of 130 volatile compounds were finally identified by matching with the NIST library and comparing the retention indexes (RIs), and the results are presented in Appendix A. These compounds include alcohols (22%), aldehydes (12.1%), ketones (18.2%), alkenes (14.4%), esters (8.3%), acids (8.3%), pyrroles (3.8%), and others (12.9%), as illustrated in Figure 1B. In terms of the number of detected compounds, alcohols, ketones, and alkenes are the top three classes of compounds in RAPT. Figure 1A shows the compounds identified in each year and the years in which those compounds were present. Of them, 64 compounds (approximately 49.23%) were shared by all RAPT samples regardless the years of storage. Furthermore, from Appendix A and Figure 1A, it is worth noting that a few of the compounds, such as 1,2,3-trimethoxybenzene, hexanal, and 2-hexenal, were produced with a storage time of over two years.

As illustrated in Figure 1 and Appendix A, the composition and contents of volatile compounds in RAPT underwent apparent changes during storage. At the same years of storage, the difference in contents between compounds was also significant (*p* < 0.05). As shown in Figure 1C and Appendix A, alcohol compounds are the most abundant component, which is independent from the years of storage. In a previous study, most of the alcohols in tea, such as benzyl alcohol, phenylethyl alcohol, and (Z)-3,7-dimethyl-2,6-octadien-1-ol, contribute to the rose-like and citrus-like aroma characteristics [9]. These alcohols are produced from the degradation of Strecker aldehydes or the hydrolysis of glycoside precursors [27]. In addition to alcohols, aldehydes, ketones, alkenes, and esters also showed relatively high contents in RAPT. The total concentration of volatiles generally fluctuated during storage (Figure 1C). Moreover, after 5 years of storage, the total volatile contents decreased, and although the overall content remains relatively stable, it is still lower compared with that in the samples with shorter storage durations. This distinction may be attributed to the differences between RAPT and other tea processing techniques.

To investigate the variation in the volatile profile of RAPT samples stored for different durations, this work conducted a principal component analysis (PCA) and a hierarchical cluster analysis (HCA) using the semi-quantitative results of 64 volatile compounds. The PCA scoring plot (Figure 1D) demonstrates a clear differentiation of the samples based on the first two principal components, which collectively account for 65.3% of the total variance. The RAPT samples can be broadly categorized into four distinct groups, indicating significant changes in the volatile profile during storage. The results of the HCA and PCA revealed a clear classification of the samples into four categories: T0, T1, T2, and T5 formed one category; T3 and T4 formed another category; T6, T7, T8, and T9 constituted a third category; and T10 represented a separate category (Figure 1D,E). Notably, the samples with a storage duration of 10 years (T10) exhibited a distinct separation from other samples, suggesting a significant change in the volatile composition following a decade of storage. In addition, the samples with a storage duration of 1–5 years were predominantly located in the left region of the scoring plot. Overall, the volatile profile of RAPT may experience a significant shift after 3–5 years of storage, representing the first turning point, followed by another notable transformation after 10 years of storage, potentially signifying the second turning point. The loading plot (Appendix A) indicated that volatile compounds underwent significant changes during the storage period, particularly in linalool, α-terpineol, 3,7-dimethyl-1,5,7-octatrien-3-ol, geraniol, cis-5-ethenyltetrahydro-α,α,5-trimethyl-2-furanmethanol, and (R)-5,6,7,7a-tetrahydro-4,4,7a-trimethyl-2(4H)-benzofuranone. These results further demonstrated the non-negligible influence of storage on the volatile composition of RAPT.

### 3.2. Dynamic Changes in Different Compounds during Storage

As indicated by the quantitative results, the contents of many volatile compounds varied greatly during the storage of RATP. To visually illustrate the change patterns of volatile compounds in RAPT for different storage years, this study employed a heatmap analysis, as depicted in Figure 2. 

As illustrated in Figure 2, three distinct patterns of changes can be observed in the content of these compounds. The first pattern consists of a gradual decrease in content over time, the second pattern involves an increase in content with increasing storage time, and the third pattern exhibits an initial increase followed by a gradual decrease after 5 years. The first group of volatile compounds includes 1-octen-3-ol, 3,5,5-trimethy-2-hexene, (E)-2-octen-1-ol, 1-hexanol, 2-methl-butanoic acid, 3-methyl-butanoic acid, and 6,10-dimethyl-5,9-undecadien-2-one. These seven volatile compounds showed a tendency to decrease in content during storage, and their dominant aroma characteristics were described as mushroom-like, green, roasted, fatty, and cheese-like. It is worth noting that most of these compounds belong to the alcohol chemical class, which is an interesting finding. Alcohols have been found to be the main volatile compounds in RAPT and contribute significantly to its characteristic aroma [10]. Thus, it is hypothesized that tea with shorter storage times may have stronger grassy, roasted, and fatty aromas. In addition, the second group of compounds includes α-pinene, D-limonene, 2-ethyl-1-hexanol, and DL-menthol. These volatile compounds are mainly associated with a woody and mint-like aroma. This finding explains the disappearance of green and roasted aroma, and the emergence of woody and mint-like aroma in RAPT. Furthermore, most of these compounds showed a pattern towards the third group of compounds (approximately 75%). For instance, benzaldehyde increased from 14.77 μg/100 g (stored for 0 years) to 46.78 μg/100 g (stored for 5 years) and then slowly decreased. Similarly, 2,4-Di-tert-butylphenol (16.86–35.94) increased from 16.86 to 35.94 (μg/100 g) initially and then slowly decreased. Here, benzaldehyde is produced through the deamination of amino acids by aminotransferase [28]. Interestingly, a majority of the third group of volatile compounds decreased in 5 years of storage (T5), which is consistent with a previous study indicating that five years of storage is a significant turning point [29]. Moreover, most of the third group of volatile compounds contributes to the floral and fruity aroma. This result suggested that the green and roasted aromas diminish over time, while the floral and fruity aromas gradually become more prominent during the 0–5 years of storage.

Appendix A shows a decrease in the relative contents of 12 volatile compounds during storage. These compounds include 1-hexanol, 2,6-dimethy-5-heptenal, 1-octen-3-ol, 2,3-octanedione, 2,6,6-trimethyl-1,3-cyclohexadiene-1-carboxaldehyde, 3,5,5-trimethyl-2-hexene, (E)-2-octen-1-ol, 2-methyl-butanoic acid, 1-ethyl-1H-pyrrole-2-carboxaldehyde, 6,10-dimethyl-5,9-undecadien-2-one, trans-β-ionone, and 6-undecanone. Specifically, they have predominantly green, fatty, floral, fruity, and roasted aroma. Moreover, the relative contents of some volatile compounds have been increased during storage, notably α-ionone, isophorone, 1,2-dimethoxy-benzene, and 1,2,5,5-trimethyl-1,3-cyclopentadiene. These compounds contribute to violet-like, woody, fruity, vanilla, and phenolic-like aromas. It is speculated that these changes in volatile compounds may explain the development of a woody aroma in RAPT with extended storage periods.

### 3.3. Aroma Characteristics of RAPT during Storage

The odor activity value (OAV) is a quantitative measure used to assess the odor potency of a compound or mixture and is typically determined by measuring the concentration of a volatile compound in a sample and by comparing it to a corresponding odor threshold. Based on the OAV, the contribution of each volatile compound can be evaluated, and the correlation between the quantified volatile compounds and the aroma can be determined. In general, aroma compounds with OAV greater than or equal to one are considered as active aromatic compounds and are believed to significantly contribute to the overall aroma profile of tea [6,30]. It is also important to consider the relative contribution of the components in a mixture, in addition to assessing the contribution of individual compounds. Therefore, the contribution of the mixed compounds to the aroma was further analyzed by calculating the aroma contribution index (ACI). Based on an OAV of ≥1, forty-three volatile compounds were identified as potential aroma compounds (Appendix A). Figure 3 shows the change in these 43 volatile compounds during storage. Interestingly, most of the OVAs had their highest values at T3 and T4, and T4 had 22. In addition, according to their different aroma characteristics, the 43 volatiles were classified into nine groups: green, fruity, floral, roasted, fatty, chemical, cheese-like, stale, and woody. These aroma attributes have been previously used to describe the aroma characteristics of RAPT in other studies.

Appendix A shows that the aroma compounds in RAPT decreased with increasing storage time, including 1-octen-3-ol, 2-methyl-butanoic acid, and 1-ethyl-1H-pyrrole. These compounds are related to green and roasted aroma attributes. On the one hand, linalool (OAV = 8040.2) and 1-octanol (OAV = 620.8) exhibited the highest OAVs at T3 (3 years of storage). (E,E)-2,4-heptadienal (OAV = 5892) and octanal (OAV = 4131.4) had the highest OAVs at T4 (4 years of storage). On the other hand, DL-menthol (OAV = 1.8) reached its maximum value at T10 (10 years of storage) and was characterized by a peppermint and woody aroma. This could explain the formation of woody and mint-like aroma in RAPT after 10 years of storage. Moreover, (Z)-3,7-dimethyl-2,6-octadien-1-ol was found to contribute to the aroma profile of RAPT during storage. Its OAVs gradually increase from 2.7 at T0 to a peak value (6.2) at T3, before slowly decreasing with increasing storage time to 2.7 at T10. Furthermore, it is interesting to note that α-ionone (OAV = 562.5) and linalool (OAV = 8040.2) have violet-like, citrus-like, and flowery aroma characteristics, reaching their peak levels at T3. Meantime, geraniol (OAV = 258.3), octanal (OAV = 4131.4), and α-terpineol (OAV = 491.5) reached the highest at T4, displaying rose-like, citrus-like, and flowery aroma characteristics. This suggests that the storage of RAPT for about 3 to 4 years results in an abundance of floral and fruity aroma. Moreover, the contribution of aroma compounds in the mixture was compared using the aroma contribution index (ACI). Five volatile compounds were screened based on their high OAVs (≥100) and ACIs (≥1). These compounds include linalool (OAV: 2691.1–8040.2, ACI: 0.72–8.28), (E,E)-2,4-heptadienal (OAV: 3248.6–5892, ACI: 0.55–7.76), (Z)-3-hexen-1-ol (OAV: 0–151,243.8, ACI: 0–40.33), 2,6,10,10-tetramethyl-1-oxaspiro [4.5]dec-6-ene (OAV: 0–345,510, ACI: 0–76.48), and octanal (OAV: 1320.7–4131.4, ACI: 0.25–3.52).

### 3.4. Identification of the Key Aroma Characteristics of RAPT during Storage

Floral, fruity, and green aromas are considered favorable aroma characteristics in tea. Therefore, we analyzed the floral and fruity aromas of volatile compounds in RAPT. In Figure 4A, it is shown that most of the volatile compounds in RAPT have pleasant floral and fruity aromas during storage. Previous results have also indicated that the floral and fruity aromas increased over time during the first four years of storage. RAPT is unique compared with other teas because it undergoes autoclaving and compressing treatments. These treatments result in an increase in the content of certain aroma compounds such as phytol, linalool, and geraniol, which have a significant contribution to the floral aroma of tea [31]. Consequently, RAPT has a distinctive floral and fruity aroma, which may be dramatically changed during storage. As shown in Figure 4B, the roasted aroma, particularly that of 1-ethyl-1H-pyrrole, gradually diminished over the storage period. After ten years of storage, the green aroma decreased, while the woody aroma increased. However, it is important to note that the floral and fruity aromas remained dominant, indicating their overall persistence in the RAPT.

In addition, the intensity of the floral and fruity were gradually enhanced during storage. Hence, eight aroma compounds were screened with OAV ≥ 100, namely linalool (OAV: 2691.01–8040.2), α-terpineol (OAV: 250.8–491.5), geraniol (OAV: 99.5–258.3), trans-β-ionone (OAV: 40730.3–108493), α-ionone (OAV: 274.1–562.5), (E,E)-2,4-heptadienal (OAV: 3248.6–5892), 1-octanol (OAV: 95.2–620.8), and octanal (OAV: 1320.7–4131.4), as illustrated in Appendix A. Interestingly, all of them contribute to the aroma of floral and fruity. Notably, (E,E)-2,4-heptadienal is one of the dominant volatile compounds in high-grade Jingshan green tea, and it is produced through the degradation of linolenic acid, providing flowery aroma [32]. In oolong tea (*Camellia sinensis*), geraniol and linalool are the main contributors to the floral, rose-like, and sweet aroma characteristics. In addition, Yang et al. [33] found that trans-β-ionone contributed to the aroma characteristics of Rougui tea. Therefore, these eight volatile compounds are considered to be the key contributors to the floral and fruity characteristics during RAPT storage.

### 3.5. Metabolic Pathways of the Key Aaroma Compounds

The changes in aroma characteristics may be influenced by several factors during storage. Geraniol and linalool, which are monoterpenes, serve as the primary constituents responsible for the pleasant floral and fruity scent found in plants. These compounds are synthesized in tea through the metabolic conversion of two primary precursors, namely isopentenyl diphosphate (IPP) and dimethylallyl pyrophosphate (DMAPP). The mevalonate pathway (MVA) plays a critical role in the synthesis of secondary metabolites, including sterols, sesquiterpenes, and triterpenes within the cytoplasm and endoplasmic reticulum. The 2-C-methyl-D-erythritol-4-phosphate pathway (MEP) and carotenoid cleavage dioxygenase (CCD) are two key metabolic pathways involved in the biosynthesis of monoterpenes and oxidation of carotenoids, respectively [34]. It is hypothesized that changes in the metabolism of microorganisms during storage may result in changes in aroma characteristics. Trans-β-ionone and α-ionone are secondary oxidation products derived from carotenoids, including β-carotene, α-carotene, phytoene, lutein, and lycopene, which undergo oxidation by CCD to form primary oxidation products and then are further converted into secondary oxidation products [28]. Furthermore, the conversion of linolenic acid into (E,E)-2,4-heptadienal occurs through the involvement of leucoanthocyanidin reductase (LOX) and glutathione peroxidase (HPL). Previous work has shown that LOX continues to facilitate the synthesis of aldehydes from fatty acids throughout the fermentation stage [35]. The storage of RAPT, in addition to being a natural aging and fermentation process, is also believed to contribute to the changes in aroma. These changes may be attributed to microbial metabolism, enzymatic digestion, and natural oxidation.

## 4. Conclusions

In the present study, a method combining HS-SPME with GC-MS was used to profile the volatile compounds in RAPT stored for different years (0–10 years). In total, 130 volatile compounds including alcohols, aldehydes, ketones, alkenes, and esters were identified, and 64 of them were shared by all tea samples. Alcohols, ketones, and alkenes were determined as the top three classes of volatiles in RAPT. It was revealed that storage has a significant effect on the volatile composition of RAPT. A notable transformation was observed after 3–4 years of storage, signifying the first turning point. Then, the volatile profile of RAPT with a storage duration of 10 years (T10) experienced a distinct separation from other samples; the likely dramatic shift after 10 years of storage may signify the second turning point. A total of 43 volatile compounds were selected based on their OAVs and aroma profiles, and most of them contributed to the floral and fruity aromas. These floral and fruity compounds displayed a pattern of initial increase, followed by subsequent decrease throughout the storage period. In contrast, the roasted aroma, especially from 1-ethyl-1H-pyrrole, was found to diminish during the storage period. Linalool, (E,E)-2,4-heptadienal, (Z)-3-hexen-1-ol, 2,6,10,10-tetramethyl-1-oxaspiro [4.5]dec-6-ene, and octanal (OAVs of ≥100 and ACIs of ≥1) were identified as the primary contributors to the aroma profile. Ultimately, eight key compounds, namely linalool, α-terpineol, geraniol, trans-β-ionone, α-ionone, (E,E)-2,4-heptadienal, 1-octanol, and octanal, were selected based on their association with floral and fruity aromas. The results of our study enhance comprehension regarding the composition of volatile compounds in RAPT over varying storage durations. However, the changes in volatile compounds were limited by the samples and storage environment, and the effects of different storage environments need to be further studied. In addition, enzymes and microorganisms can be analyzed to explore the pathways and mechanisms of conversion between substances during storage.

## Figures and Tables

**Figure 1 foods-12-03568-f001:**
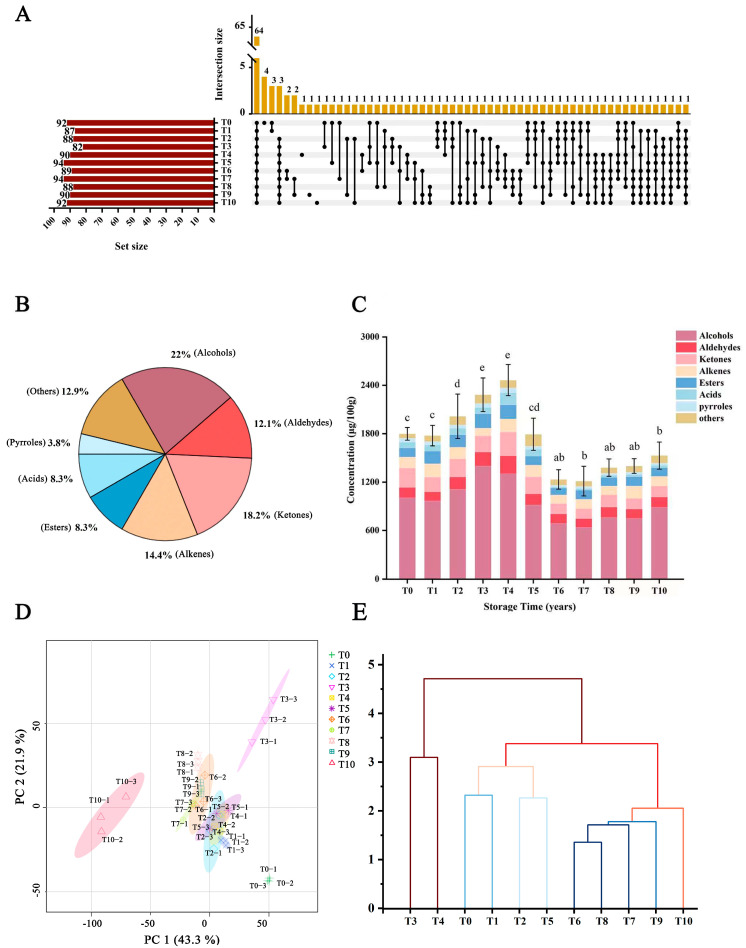
Annotation of volatile compounds during RAPT storage: (**A**) upset diagram; (**B**) pie diagram. The proportion of different types of volatile compounds in RAPT: (**C**) total volatile compound content and classification. (**D**) Principal component analysis (PCA). (**E**) Hierarchical cluster analysis (HCA).

**Figure 2 foods-12-03568-f002:**
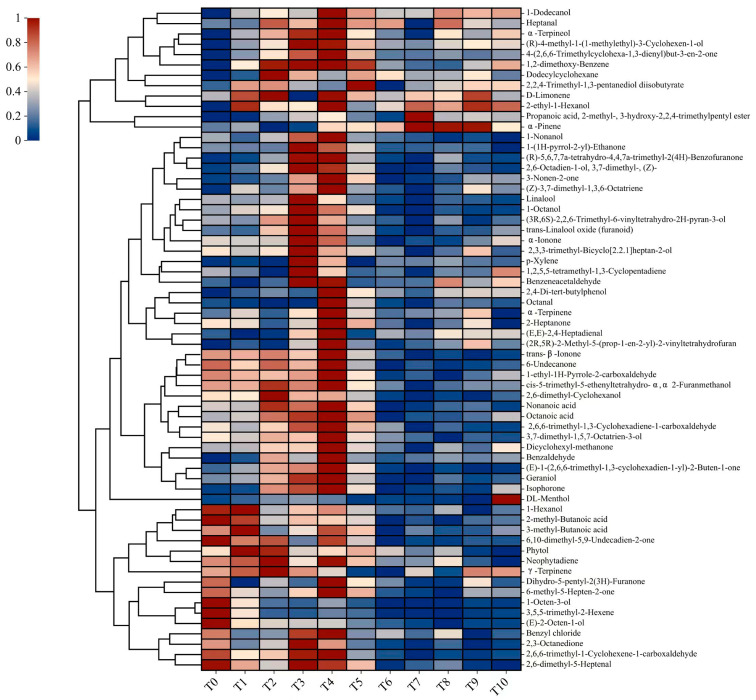
Heat map of differential volatile compounds in the samples of RAPT at various storage time points.

**Figure 3 foods-12-03568-f003:**
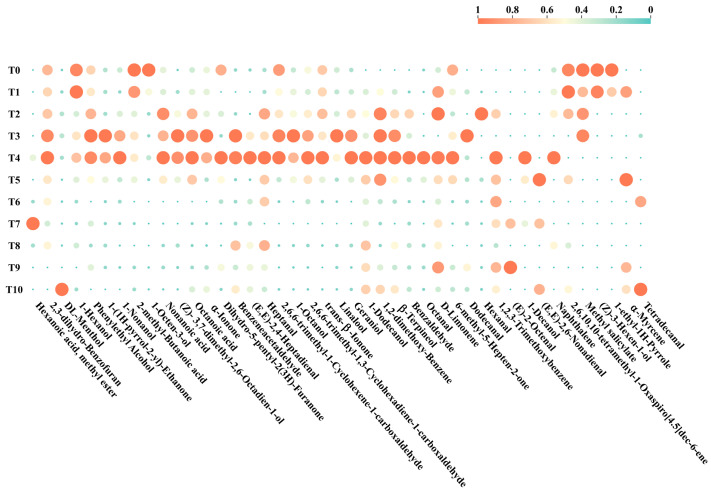
Based on OAV > 1 analysis of 43 volatile compounds and during RAPT storage.

**Figure 4 foods-12-03568-f004:**
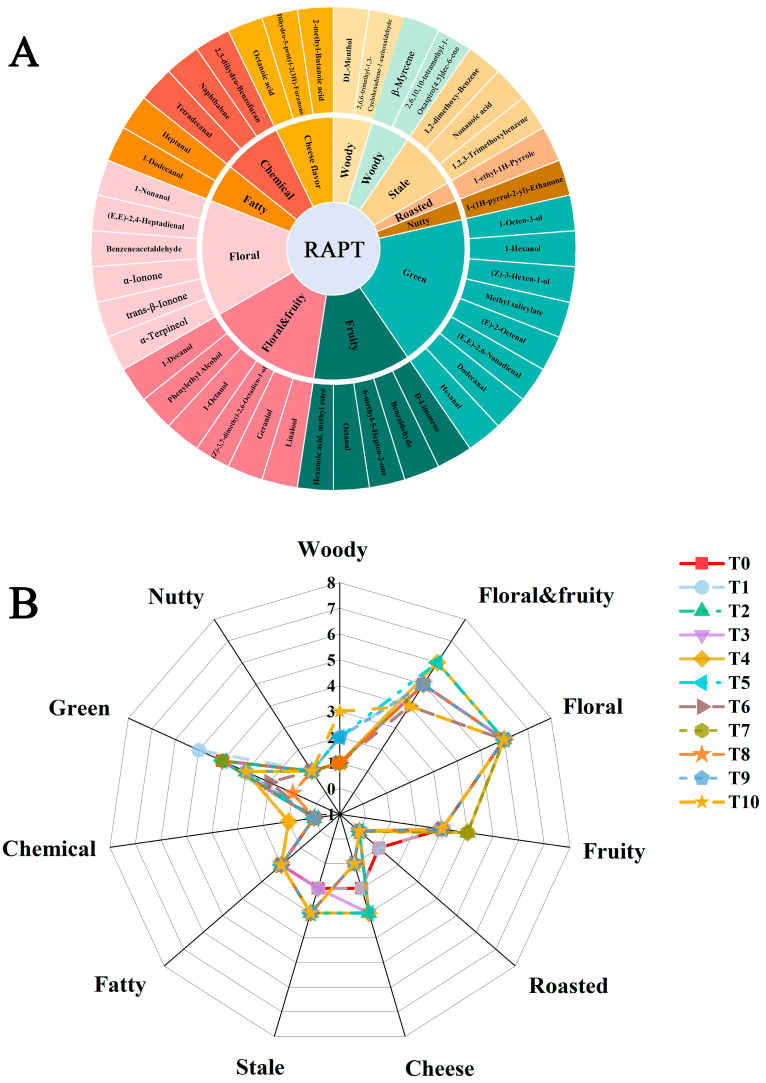
Based on OAV analysis of volatile compounds during RAPT storage. (**A**) Flavor wheel of 43 key volatile compounds in samples at different storage times. (OAV > 1). (**B**) Radar plots of flavor characteristics of RAPT during storage. (OAV > 1).

## Data Availability

The data are unavailable due to privacy or ethical restrictions.

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
