# Peer review of "Analysis of Volatile Profile and Aromatic Characteristics of Raw Pu-erh Tea during Storage Based on GC-MS and Odor Activity Value"

_foods, 2023, doi:10.3390/foods12193568_

Round 1

Reviewer 1 Report

The title is too lengthy; the authors attempted to encapsulate their manuscript's entire summary within it. It is advisable to condense it to a title of 12 to 15 words, no more. Furthermore, it is essential to determine if 'volatiles' and 'aromatics' could be considered synonymous. The three treatments should also be emphasized, and the technique can be streamlined to GC-MS.

The abstract is clear and well-structured.

Please keep in mind that keywords are the words that will help you search for your manuscript in databases and help search engines. To the extent possible, try to avoid repeating them in the title.

Is it a Pu-erh tea variety of Camellia sinensis? Clarify in the introduction. 

Line 63. In the reference for Fan et al., please include the year.

Line 65. In the reference for Qi et al., please include the year. And review all the document. 

The introduction is clear and concise, effectively presenting the authors' research question and the objective of their manuscript.

Line 191. In this section, it should be clarified: How many years were the samples stored, or at the very least, provide a range of storage duration.

Please do not refer to "see Tables 1, 2, etc." in the manuscript if these tables are in the supplementary materials. Kindly indicate that they are in this section every time they are referenced in the text.

For your next submission, please include the supplementary material as a PDF. The tables have shifted when I downloaded the file.

The results section is clear, with each of the figures thoroughly described in the text and accompanied by a substantial discussion of the relevant literature. The manuscript's discussion is robust.

The conclusions presented by the authors are excessively long; they once again summarize their results. In this regard, I recommend only providing relevant conclusions, especially regarding the variation profile over the 11 years, precisely the volatile compounds that could be attributed as chemical markers. Lastly, the authors should outline their work's perspective and future directions.

In general, the article presented by the authors is interesting and aligns with the journal's scope. Overall, my comments have been few because I believe the manuscript meets the necessary scientific rigor for publication. However, this time, due to the details in the title and conclusions, I will make major revisions. This is to review these points again and determine whether the article can be published if they are addressed as requested.

Reviewer 2 Report

1.     I propose to clarify the title of the article, it is too long, poorly specified.

2.     Please complete the abstract with the research results obtained with more details.

3.     What is 7542 series, please explain - line 113-114.

4.     Please provide a morphological description of the harvest phase and sampling method.

5.     Figure B or Figure ! B – line 202.

6.     The data quoted in the text - lines 205-206, are not presented in Fig. 1A, as it would appear from the content.

7.     The different types of alcohols are not shown in Figure 1-201-217.

8.     Please improve Fig. 1, 1A - put more information, 1 D - improve readability.

9.     Read the text carefully, correct punctuation and characters.

10.  If the highest OAV is at T3 or T4 than not after (after 3 or 4 years of storage) - line 308, next if 22 volatile compounds is true at T3 and T4?

11.  Please correct the caption Fig. 4 - compounds and during.

12.  In Fig. 4B, it is difficult to trace the trends of parameter changes, especially at T10.

13.  Data presented in the text - lines 359-36, please refer to the table.

14.  Stored time is 0-10 or 0-11- line 394

Round 2

Reviewer 1 Report

The authors addressed each of my comments and provided a clear response to each of them. They have improved the title, keywords, and their conclusion. For now, as a minor comment, I recommend reviewing the resolution of each of the figures in detail, ensuring that they are at 300 dpi. Finally, I recommend that the manuscript be published.